# A Micro-Nano Structure Formed by SiC/Graphene Oxide Self-Assembly Improves the Wear Resistance and Corrosion Resistance of an Epoxy-Based Composite Coating

**DOI:** 10.3390/polym14214704

**Published:** 2022-11-03

**Authors:** Jun Tang, Yali Tan, Fugang Qi, Biao Zhang, Ao Zhou, Nie Zhao, Xiaoping Ouyang

**Affiliations:** 1School of Materials Science and Engineering, Xiangtan University, Xiangtan 411105, China; 2Key Laboratory of Low Dimensional Materials and Application Technology of Ministry of Education, Xiangtan University, Xiangtan 411105, China; 3Hunan Bangzer Technology Co., Ltd., Xiangtan 411100, China

**Keywords:** corrosion resistance, tribological properties, composite material, organic coatings, interface regulation

## Abstract

In air and railway transportation, corrosion and wear lead to the rapid failure of equipment. Epoxy (EP)-based coatings are widely used in research on the anti-corrosion of organic coatings, but their application as materials for wear resistance is limited due to their non-abrasive properties. In this study, a novel high-performance epoxy-based composite coating was developed through the self-assembly of silicon carbide (SiC) and graphene oxide (GO) and the tuning of the interfacial structure with epoxy resin. The coatings were comprehensively characterized, including their electrochemical behavior, a salt spray test, and friction and wear experiments, and the optimal addition ratio of SiC-G@GO (SiC-G@GO was prepared by the self-assembly of γ-(2,3-epoxypropoxy) propytrimethoxysilane (KH560)-modified SiC (SiC-G) on the surface of GO sheets) in the epoxy coatings was explored. Benefiting from the labyrinth effect and their rolling-friction-like microstructure, the 1 wt% SiC-G@GO/EP coating exhibits excellent wear and corrosion resistance. Compared with pure epoxy resin, the 1 wt% SiC-G@GO/EP coating increased by 4 orders of magnitude after 10 days of immersion. The average friction coefficient was 41.5% lower than that of the pure EP coating, and the wear rate was 56.6% lower. This research has positive implications for the development and application of anti-corrosion and wear-resistant organic coatings.

## 1. Introduction

Due to their strong adhesion and small curing shrinkage, epoxy-based coatings account for about 60% of all anti-corrosion coatings [1]. However, epoxy resins are brittle and have poor mechanical properties, and the coatings are easily damaged. Their destruction produces defects, and corrosive substances such as oxygen, water molecules, and ions easily reach the surface of the substrate through cracks and holes, causing corrosion, which is not conducive to long-term corrosion protection [2,3,4]. Many studies have shown that adding nanofillers to organic coatings can effectively fill the micropores so as to block the penetration of corrosive substances, thereby prolonging the service life of the coating. Examples of these fillers include: Al_2_O_3_ [5,6], CeO_2_ [7], ZrO_2_ [4], ZnO [8], SiO_2_ [9,10], TiO_2_ [11], Fe_3_O_4_ [12], SnO_2_ [13], hexagonal boron nitride (h-BN) [14,15], molybdenum disulfide (MoS_2_) [16], graphene oxide (GO) [17], etc. F.X.qin et al. found that MoS_2_ and graphene have excellent shielding effects on hybrid composites due to their large diameter thickness ratio and orderly arrangement of fillers [18,19,20]. These fillers can effectively improve the barrier effect of organic coatings. In particular, a series of sheet materials, such as graphene and its derivatives, are good candidates for anti-corrosion coatings due to their fascinating chemical and thermal stability and impermeability [21]. However, these anti-corrosion studies have ignored the wear resistance of the material. In the fields of aviation and rail transportation, parts and components are inevitably affected by wear, corrosion, and other damage, which requires the coating to have not only a good corrosion resistance but also excellent wear resistance, because friction can destroy the surface coating of the substrate. The layer structure forms defects, and the corrosive medium easily invades through these defects, thereby causing corrosion [1,22,23]. However, there are few studies on epoxy-based coatings that are both anti-corrosive and wear-resistant.

In recent years, SiC has been able to meet the requirements for wear-resistant coatings due to its excellent physicochemical properties and excellent wear resistance [24]. Sunny Chung et al. found that the wear resistance of composites increased with the increase in the SiC content [25]. Qiang Li et al. used a laser multi-pass overlapping process to clad a mixture of 30 vol.% SiC and 70 vol.% Ni-based alloy onto a steel substrate to reduce the friction coefficient [26]. However, due to the high specific surface energy of SiC, it easily agglomerates [27], which limits its application to organic coatings, because the direct addition of SiC not only reduces the corrosion resistance of the coating but also reduces the adhesion of the coating [28]. Compared with the larger diameter-to-thickness ratio of GO, the smaller volume of SiC, when used alone, has a poorer pore-filling effect on epoxy coatings, which limits the improvement of the coating performance.

Using GO as a carrier for SiC in the epoxy coating can enhance the filling of defects of the coating [29] but, on the other hand, it can also convert the abrasive friction of SiC in the coating into the friction of SiC on the surface of GO, thereby improving the wear resistance of the coating. For GO, the coating of SiC on its surface can increase the insulation and dispersibility of GO [30] and improve the dispersibility of SiC, while improving the compatibility of SiC with epoxy resin. Therefore, it is concluded that SiC has a good wear resistance but poor anti-corrosion performance, and graphene has an excellent anti-corrosion performance but insufficient wear resistance. The combination of the two can complement each other. SiC has functional groups that can be used for grafting after modification with a coupling agent, and there are abundant functional groups on the surface of GO [29,31]. These conditions provide the possibility of the self-assembly of SiC and GO.

Based on the above analysis, we developed a new strategy with which to fabricate nanocomposite coatings, and SiC-G@GO was prepared by the self-assembly of γ-(2,3-epoxypropoxy) propytrimethoxysilane (KH560)-modified SiC (SiC-G) on the surface of GO sheets (). Using the impermeability of the two-dimensional material, graphene oxide, to form a labyrinth effect (Figure 1) in the epoxy coating so as to increase the diffusion path of the corrosive medium [32], the lamellar structure of GO and spherical silicon carbide form a microstructure similar to the rolling friction observed in the epoxy resin. Electrochemical impedance spectroscopy (EIS) and salt spray tests were carried out on the composite coating, which confirmed that the modulus value of the composite coating with SiC-G@GO increased by 3 to 4 orders of magnitude at a low frequency compared with the pure EP coating. The friction and wear experiments show that the wear resistance of the coating is improved under the normal load (30 N), the average friction coefficient is reduced by 41.5% compared with the neat EP coating, and the wear rate is reduced by 56.6%. This research has important reference value for the development of anti-corrosive and wear-resistant organic coatings and provides new research ideas for wear-resistant coatings.

## 2. Experimental Program

This section describes the experimental methods used to characterize the effectiveness of the incorporation of nanomaterials (SiC-G@GO) into the coating, in which the epoxy was selected as a prime as a representative. The detailed information on the materials, dispersion methods, and the procedure of the experiments are described below.

### 2.1. Materials and Sample Preparation

#### 2.1.1. Materials

Graphite, 98% concentrated sulfuric acid (H_2_SO_4_), potassium permanganate (KMnO_4_), sodium nitrate (NaNO_3_), 30% hydrogen peroxide (H_2_O_2_), absolute ethanol (analytical grade), 37 % hydrochloric acid (HCl), sodium chloride (NaCl), and 3-glycidyloxypropyltrimethoxysilane (GLYMO) were purchased from Sinopharm Holding China Chemical Reagent Co., Ltd. (Shanghai, China). Nano-silicon carbide (40 nm) was purchased from Aladdin Reagent (China) Co., Ltd. (Shanghai, China), and epoxy resin (CYD 014) and its hardener (P54) were provided by Baling Chemical (China) Co., Ltd. (Yueyang, China)). Q235 steel plates were purchased from Guangzhou Biugeda (China) Co., Ltd. (Guangzhou, China).

GO was synthesized by the oxidation of graphite powder with sulfuric acid and potassium permanganate (H_2_SO_4_-KMnO_4_) (Hummers’ method).

#### 2.1.2. Preparation of SiC-GO Composite Filler

We dispersed 0.8 g of SiC in 200 mL of ethanol containing 1 mL of GLYMO and 1 mL of acetic acid. The suspension was sonicated for 1 h, stirred at 70 °C for 4 h, and finally washed with ethanol to remove excess HFTMS. It was then dried and supplemented with 200 mL of absolute ethanol with 0.8 g GO, and then stirred for 4 h. After centrifugation and drying for 24 h, the composite filler SiC-G@GO was obtained. Figure 2 is a schematic diagram of the preparation of the SiC-G@GO composite filler.

#### 2.1.3. Preparation of the Coating

The coatings were prepared by adding the GO, SiC, and SiC-G@GO materials to epoxy resin (CYD 014), followed by mechanical stirring for 2 h and ultrasonic shaking for 0.5 h. In order to cure the coating smoothly, a certain proportion of curing agent was added to the abovementioned composite coating, and then it was mechanically stirred until uniform. The amount of curing agent added was calculated according to the epoxy value of the coating and the amount of active hydrogen in the curing agent. Subsequently, the obtained coating was sprayed on the Q235 low-carbon steel surface and an iron block (20 mm × 20 mm × 5 mm), which was sandblasted and ultrasonically cleaned with absolute ethanol. All the samples were cured at room temperature for 48 h, and the thickness of the composite coating sprayed on the mild steel plate was 90 ± 5 μm. SiC-G@GO_1_ was an EP coating with 1 wt% SiC-G@GO, and SiC-G@GO_2_ was an EP coating with 2 wt% SiC-G@GO.

### 2.2. Sample Characterization

Fourier transform infrared (FTIR) spectroscopy was performed using a Nicolet 6700 spectrometer in the range of 400–4000 cm^−1^, X-ray diffraction (XRD, D/MAX-2500/PC, Japan), X-ray photoelectron spectroscopy (XPS, Thermo Scientific K-Alpha, Waltham, MA, USA), a field emission scanning electron microscope (FE-SEM, SU5000, Tokyo, Japan) and transmission electron microscope (TEM, JEM-2100, Tokyo, Japan) to characterize the morphologies of the composites. Wear scars were observed with a super-depth-of-field three-dimensional microscope system (VHX-2000C, Osaka, Japan) and a field emission scanning electron microscope (TESCAN MIRA4, Brno, Czech Republic).

### 2.3. Electrochemical Characterization and Long-Term Performance of the New Composite Coatings 

#### 2.3.1. EIS Test and Accelerated Durability Test

The salt spray test was carried out in a salt spray test chamber (BDG 882, Guangzhou, China) with a salt concentration of 5%, and the scratched samples were exposed to the salt spray for 20 d, with an exposure area of 120 mm × 50 mm. The anticorrosion properties of the composites were carried out on an electrochemical workstation (Gamry reference 600+, Philadelphia, PA, USA). The anticorrosion properties of the coatings in 3.5% NaCl aqueous solution were investigated by open circuit potential (OCP) and electrochemical impedance spectroscopy (EIS). Electrochemical measurements were performed using a conventional three-electrode system with a platinum sheet counter electrode, saturated calomel reference electrode (SCE), and working electrode (1 cm^2^ exposed area). EIS measurements were performed at various immersion times in the frequency range of 100 kHz–10 MHz and a sinusoidal voltage with an amplitude of 10 mV. We began taking potentiodynamic measurements from a potential of −400 mV to +400 mV near the OCP, with a scan rate of 1 mV/s. All impedance measurements were performed using a Faraday cage to minimize external disturbances, and the experimental data were fitted using ZsimDemo software.

#### 2.3.2. Evaluation of Corrosion Resistance in Short/Long Runs

The EIS data were further analyzed using ZsimpWin software. The equivalent circuit diagrams used to fit the EIS data (fitting error < 1 × 10^−4^) are shown in Figure 3 and Figure 4. Figure 3 shows the corresponding equivalent circuits of EP, GO, and SiC. Figure 4 is an equivalent circuit diagram of the SiC-G@GO nanocomposite coating. In the early stage of immersion, the electrolyte has not penetrated the coating, and the coating is equivalent to a large resistance. In the middle stage of immersion, the electrolyte diffuses in the coating through the micropores generated by the immersion, resulting in diffusion resistance. In the later stage of the coating immersion, with the formation of macro-pores, the concentration gradient originally present in the organic coating disappears, and new resistance is formed in the interface area due to the corrosion products of the base metal.

R_s_ is the solution resistance, Q_c_ is the interface capacitance between the solution and the coating, R_c_ is the resistance of the non-invaded coating, and R_d_ is the diffusion resistance after the intrusion of the corrosive liquid. R_d1_ is the resistance of the part with a higher intrusion concentration of the corrosive liquid, and R_d2_ is the coating resistance with low invasion concentration of corrosive liquid, Q_d_ is the interface capacitance between the invasion and non-intrusion of the corrosive solution, Q_d1_ is the interface capacitance between the part with a higher concentration of the corrosive solution and the part with a lower concentration, and Q_d2_ is the intrusion of the corrosive solution. With respect to the interfacial capacitance, R_ct_ is the coating that the corrosive liquid has not yet penetrated. Q_i_ is the interface capacitance between the base metal and the etching solution, and R_i_ is the resistance of the corrosion product (Figure 3 and Figure 4).

The circuit diagrams fitted by the EIS data for the EP, GO, and SiC coatings are explained as follows:

For pure epoxy coatings, due to their relatively low degree of curing, it is easier to form channels for corrosive media micropores. Therefore, after immersion for 0.5 h, the corrosive medium only slightly invaded the coating, and the diffusion time in the coating was short, and substrate corrosion was detected on the 10th day. For the EP coating with 1 wt% GO and SiC filler, the corrosion on the 10th day was not as substantial as that of the pure EP coating, which was roughly judged from the EIS pattern. This is because GO and SiC have a poor dispersion in EP and cannot achieve the expected effect. 

The circuit diagram fitted by the EIS data on the SiC-G@GO composite coating is explained as follows:

For coatings with added SiC-G@GO fillers, due to the strong impermeability of GO, these fillers make the diffusion of the electrolyte more difficult, and the mass transfer process of particles participating in the interfacial corrosion reaction may occur as a slow step. The resistance R_d_ caused by the diffusion process often occurs in EIS; thus, the phenomenon of the diffusion and delamination of the corrosive medium is prone to occur in the coating [33] (Figure 5). In electrochemical measurements, in the early stage of immersion, the diffusion has a steady state. That is, the mass concentration does not change with time, which can be described by Fick’s first law of diffusion (1). In coatings with strong barrier properties and large diffusion distances, there is a concentration gradient, and the concentration at certain points changes with time, which is consistent with Fick’s second law of diffusion (2) [34].
(1)$$J=-D\frac{d\rho }{dx}\;$$
(2)$$\frac{\partial \rho }{\partial t}=\frac{\partial }{\partial x}\left(D\frac{d\rho }{dx}\right)$$

J is the diffusion flux, *D* is the diffusion coefficient, *ρ* is the mass concentration of the corrosive medium, *t* is the immersion time, *x* is the distance of the corrosive medium invading the coating, and the negative sign indicates that the diffusion direction of the substance is opposite to the direction of the mass concentration gradient *dρ*/*dx*.

It is assumed that the corrosive medium penetrates the coating through the coating pores in a steady-state process; that is, the diffusion rate of the corrosive medium on the entire interface of the coating is the same, and the corrosion depth does not change with the plane position of the coating. At the same time, it is assumed that the electrical properties of the coating on the same plane are the same before and after being eroded by the corrosive medium, and the thickness of the coating does not change with the penetration of the corrosive medium, ignoring all other external factors.

Figure 5a shows the initial soaking of the coating, where the coating is divided into non-invaded layers, and the coating is one layer. Figure 5b shows the immersion for a certain period of time. After the initial intrusion of the corrosive liquid, the coating is divided into an invading layer and a non-invading layer, so that the coating is divided into two layers. Figure 5c shows that the immersion time is longer, and the corrosive solution is more substantially diffused. It is divided into the first layer, with a high concentration of the corrosion solution, the second layer, with a low invasion concentration of the corrosion solution, and the third layer, without immersion.

Circuit diagram 1: early stage (the corrosive medium has not yet penetrated the coating). Circuit diagram 2: intermediate stage 1 (the corrosive medium’s intrusion into the coating has not yet reached the metal interface). Circuit diagram 3: mid-term 2 (the corrosion medium’s intrusion into the coating has not yet reached the metal interface but has a large concentration gradient). Circuit diagram 4: late stage (the corrosion medium’s intrusion into the interface between the coating and the metal substrate).

### 2.4. Evaluation of Wear Behavior and Abrasion Resistance

The samples were tested for their wear resistance. The friction and wear testing machine (CFT-I, Lanzhou) was used for the performance, with a vertical load of 30 N, friction back and forth at a speed of 83.3 mm/s, and the wear time of 5 min, while the friction ball was a zirconia ball with a stroke of 10 mm. The abrasion resistance was then calculated through the mass loss in the form:(3)$$\mathrm{Wear}\;\mathrm{amount}=M_{before\;experiment}-M_{after\;experiment}\;$$
where $M_{before\;experiment}$ corresponds to the weight of the initially intact coating (mg), and $M_{after\;experiment}$ refers to the remaining mass after the abrasion cycles (mg). The wear index, WR, was used to gain the input of the abrasion resistance of the new coating in the form:(4)$$\mathrm{WR}=\frac{M_{before\;experiment}-M_{after\;experiment}}{Wear\;path•Load}\;$$
where Wear path corresponds to the total length of the wear, and the unit of WR is g•N^−1^•m^−1^. The lower the wear index is, the better the abrasion resistance of the new coating is. The higher abrasion resistance can protect the coating and associated substrate against the damaging effects of the abrasive situations that are often experienced by civil structures.

## 3. Results and Discussion

### 3.1. Characterization of SiC-G@GO

FT-IR was used to study the modification of SiC and GLYMO and the grafting between SiC and GO. The FT-IR spectra of SiC, SiC-G, GO, and SiC-G@GO are shown in Figure 6a. There are three possibilities for the interaction between GO and GLYMO: the hydrolysis of the alkoxy group of GLYMO and the hydroxyl group of GO, hydrogen bonding, and the nucleophilic substitution of the epoxy and alkoxy groups. In most cases, silanes are hydrolyzed prior to surface treatment [35]. In the case of SiC-G, a broad band around 3447 cm^−1^ is observed, which can be attributed to adsorbed water, with a strong absorption peak at 829 cm^−1^, which is derived from the epoxy groups of GLYMO. The bands at 2921 cm^−1^ and 2850 cm^−1^ are attributed to the asymmetric and symmetric stretching vibrations of -CH_2_, respectively [36], which indicates that GLYMO successfully chemically bonded with the SiC powder in the form of covalent bonds. In addition, at 473 cm^−1^ and 1072 cm^−1^, we can observe the symmetric variable-angle vibrational frequencies and the antisymmetric stretching vibrational frequencies of the Si-O-Si bond, respectively [37], which indicates that GLYMO successfully modifies SiC. The hydroxyl bending vibration peak at 1383 cm^−1^ is significantly weakened [36], indicating that hydroxyl is consumed by GLYMO. Similarly, the hydroxyl bending vibration peak of GO at 1383 cm^−1^ is also significantly weakened, which indicates that the hydroxyl group of GO is reduced due to the reaction, indicating that the hydroxyl group reacts with the GLYMO molecules. These results confirmed the deposition of SiC nanoparticles on the GO surface through the reaction with KH560. XRD tests were performed on the GO, SiC, and SiC-G@GO composites, and the results are shown in Figure 6b. The GO spectrum shows a high-intensity diffraction peak near 2θ of 10.98°, which is the diffraction peak of the (001) crystal plane of graphene oxide, indicating that the crystal structure of the graphite can be transformed into the structure of graphene oxide through an oxidation reaction. In addition, GO showed a broad peak at 20–35°, which is due to the trace of unoxidized graphite remaining in GO [38]. The diffraction peaks at 34.12, 35.67, 60.27, and 71.90 in Figure 6b are characteristic peaks of SiC, corresponding to the diffraction peaks of the (111)(220)(311) crystal plane [39]. In the spectrum of the SiC-G@GO composite, the characteristic peaks of SiC still exist, which proves that the nano-SiC particles are loaded on the graphene oxide sheet. At the same time, the intensity of the SiC diffraction peaks decreases, which indicates that the crystallinity of the titanium powder in the composite material is decreased. In addition, in the spectrum of the composite material, the characteristic peak intensity of the (001) crystal plane of graphite oxide is greatly weakened, which indicates that the atomic arrangement of the graphene oxide is more disordered. The positions of the diffraction peaks shifted, to different degrees, in the direction of small angles, indicating that the distance between the lamellae increased.

XPS was further performed to characterize the elemental chemical state of the samples. Figure 6c shows the full spectra of the GO and SiC-G@GO composites. Evidently, the XPS spectrum of SiC-G@GO has two characteristic peaks originating from GLYMO at Si2p and Si2s, compared with GO, which further confirms the covalent functionalization of GO by GLYMO. The Si2p peaks of the SiC-G@GO composite can be deconvoluted into three Gaussian curves, As shown in Figure 6d, the peaks are located at 100.74, 103.25, and 102.28 eV, which are attributed to the Si-C single bond and Si-O-C bond. The appearance of the Si-O-Si bonds indicates that KH560 (GLYMO) successfully connects SiC and GO [40]. Furthermore, the C1s high-resolution spectra of GO and SiC-G@GO in Figure 6e,f provide more evidence. Compared with graphite oxide, the C1s signal of the GLYMO-modified SiC grafted with GO was significantly increased, and the appearance of C-O-Si (285.6 eV) and C-Si (283.9 eV) indicates that GLYMO molecules were grafted onto the GO surface [41].

### 3.2. FE-SEM and TEM Analyses

The SiC, GO, and SiC-G@GO materials were observed by TEM and FE-SEM. SiC and SiC-G@GO were characterized by TEM, and the results are shown in Figure 7a,b. Figure 7a shows the TEM images of SiC, and the silicon carbide has a particle size of about 40–50 nm. Figure 7b shows the TEM image of SiC-G@GO. It can be seen from the figure that SiC is grafted onto GO, and the distribution is very dense, which indicates that the coverage is relatively complete. GO is almost transparent under electron beam irradiation, mainly due to the ultrathin structure of the graphene material. SiC and SiC-G@GO were characterized by FE-SEM, and the results are shown in Figure 7c–e. Figure 7c shows the unmodified SiC nanoparticles, and it can be seen that the nanoparticles are substantially agglomerated. Figure 7d shows the FE-SEM image of the GO. It can be seen that the graphene oxide is a two-dimensional material with a large diameter thickness ratio. Figure 7e shows the SiC-G@GO composite nanomaterials. SiC is attached to the surface of GO, and there are a large number of SiC particles.

Field emission scanning electron microscopy observations were performed on the cross-sections of the coatings. The pure epoxy showed more microporous defects, as shown in Figure 8a. However, with the addition of filler, the epoxy coating of the composite filler had fewer or no micropores. This is because SiC-G@GO promotes the curing of the epoxy coating; thus, there are fewer voids, and the SiC-G@GO_0.5_ coating is shown in Figure 8b. The surface of SiC-G@GO_2_ was observed to be non-smooth, with many small particles agglomerated (Figure 8d) due to the agglomeration of the fillers, which affects the adhesion of the epoxy coatings. However, the cross-section of SiC-G@GO_1_ is almost perfect, with no holes and no small particles (Figure 8c), because the proportion of 1 wt% is moderate. This shows that the proportion of 1 wt% produces less defects.

### 3.3. Dispersion Experiment

The dispersion of the powder filler in the resin directly affects the performance and stability of the coating; thus, it is necessary to study the dispersion stability of the SiC, GO, and SiC-G@GO fillers and their dispersion in EP. It is clear from Figure 9 that all the samples were well dispersed in EP in the initial stage. However, 10 days later, SiC-G@GO was still well dispersed in EP, but GO had an obvious stratification in EP, and SiC also had stratification in EP, indicating that the pure SiC and GO have a poor dispersion in EP. As for SiC-G@GO, the dispersibility of the composites and the dispersibility of the original particles in organic solvents were significantly improved.

### 3.4. Cure Kinetic Analysis

The performance of organic composite materials depends to a large extent on the cross-linking and interfacial adhesion between the filler and polymer matrix, and the same is true for epoxy-based composite coatings. To gain insight into the curing potential of the pure EP coatings and SiC-G@GO composite coatings with different contents, we conducted a DSC thermal analysis of the non-isothermal curing process, as shown in Figure 10, with a heating rate β of 10 °C/min. In addition, curing parameters were extracted from the DSC thermograms (Table 1) and calculated by the CI, as follows [42,43]:(5)$$\Delta H^{*}=\frac{\Delta H_{c}}{\Delta H_{Ref}}\;$$
(6)$$\Delta T=T_{Onset}-T_{Endset}\;$$
(7)$$\Delta T^{*}=\frac{\Delta T_{c}}{\Delta Ref}\;$$
(8)$$\mathrm{CI}=\Delta H^{*}\times \Delta T^{*}\;$$
where Δ*Hc* and Δ*H_Ref_* are the heat released during the curing of the epoxy-based composite coating and blank EP coating, respectively, obtained from the DSC thermograms. *T_Onset_* and *T_Endset_* represent the start and end temperatures of the curing process, respectively. Δ*Tc* and Δ*T_Ref_* represent the curing temperature ranges of the composite coating and pure EP coating, respectively. 

It can be seen from Figure 10 that only one exothermic peak appears for all the coatings, which indicates that the curing reaction mechanism of the epoxy coatings is not changed by adding different kinds of nanofillers [44]. However, the composite coatings exhibited excellent curing (*ΔT* < CI < ΔH**) (as shown in Table 1). The difference in the cured states of the coatings can be explained as follows: compared with pure EP, the SiC-G@GO composite coating has a denser cross-linked network, which is attributed to the fact that SiC-G@GO contains more hydroxyl and epoxy groups. The hydroxyl (-OH) reactive groups can attack the epoxy groups in EP and facilitate the ring-opening reaction of the epoxy groups, which increases the reaction endpoints, thereby forming a denser cross-linked epoxy network. Compared with the pure EP coating and other ratio composite coatings, SiC-G@GO_1_ has a higher *ΔH* value and smaller *ΔT** value, indicating that more heat is released in a shorter time and that the cured crosslink density is higher. Meanwhile, since the SiC-G@GO composite coating has a *ΔT** < 1 compared to the EP coating, the system is very active and can be cured in a narrower temperature range [42]. This excellent curing behavior conforms with our expectations and lays the foundation for enhancing other properties of the composite coating.

### 3.5. Anti-Corrosion Performance of Composite Coatings

#### 3.5.1. EIS Measurement of Composite Coatings

EIS was further used to evaluate and compare the corrosion protection of the composite coatings. The Nyquist plots and corresponding Bode plots of the pure epoxy resin, GO/EP, and SiC/EP after soaking in 3.5 wt% NaCl aqueous solution for different times are shown in Figure 11. As shown in Figure 11a,c,e, the order of the resistance arc radius of the different coatings is SiC/EP > GO/EP > Neat EP. The size of the impedance arc radius determines the size of the impedance, and the size of the impedance reflects the resistance of the corrosion reaction, which means that by adding fillers to the pure epoxy coating, the coating’s effect on the substrate can be improved to varying degrees. In general, the impedance modulus at the lowest frequency (Z_f_ = 0.01 Hz) in the Bode plot can be defined as a parameter of the barrier properties of the coating [45,46]. As shown in Figure 11b,d,f, after soaking for 1 d, the Z_f_ = 0.01 Hz value of the neat epoxy resin is 1.25 × 10^10^ Ω cm^2^, and the Z_f_ = 0.01 Hz value drops sharply to 6.71 × 10^5^ Ω cm^2^ (Figure 11b) after soaking for 10 d. Furthermore, Figure 11b shows that the pure epoxy exhibits only two constants after 0 h of immersion, indicating the penetration of the coating/substrate interface by corrosive electrolytes (mid-immersion stage), which is due to the presence of many microscopic pores, meaning that corrosive media diffuse more easily to the coating/substrate interface [47]. For the coatings of GO and SiC, the GO coating was 8.39 × 10^10^ Ω cm^2^ at Z_f_ = 0.01 Hz (Figure 11d) and 1.04 × 10^11^ Ω cm^2^ after SiC immersion for 0 h (Figure 11f). After soaking for 10 d, the Z_f_ = 0.01 Hz values of the above two coatings are 5.21 × 10^6^ Ω cm^2^ and 9.48 × 10^6^ Ω cm^2^, respectively, which are one order of magnitude higher than those of the pure epoxy resin, which is attributed to the effect of the filler on the epoxy coating. Micropores have a certain filling effect. The above results show that the GO and SiC fillers can improve the corrosion resistance of epoxy coatings, but the improvement is limited.

For pure epoxy coatings, due to their relatively low degree of curing, it is easier for the corrosive media micropores to form channels. Therefore, after immersion for 0.5 h, the corrosive medium only slightly invaded the coating, and the diffusion time for the coating was short, and the substrate corrosion was detected on the 10th day. For the EP coating with 1 wt% GO and SiC filler, the corrosion on the 10th day was not as substantial as that of the pure EP coating, which was roughly judged from the EIS pattern. This is because GO and SiC have a poor dispersion in EP and cannot achieve the expected effect. Table 2 shows the circuit diagrams for the fitting parameters and assignments based on the EIS data.

Due to the unmodified GO, the poor dispersion of SiC in the epoxy coating fails to fully exert the performance of the filler, which is too far from the expected value. Therefore, the GO and SiC materials were modified, and the test performance is shown in Figure 12. By comparing the Nyquist plots and the corresponding Bode plots after immersion in 3.5 wt% NaCl aqueous solution for different times, the overall modified corrosion resistance of the epoxy coating is significantly improved by the filler, as shown in Figure 12. For example, the modulus value at Z_f_ = 0.01 Hz is at least 3 orders of magnitude higher than that of the pure epoxy, which is due to the good dispersion of the flakes. The layered material fills the micropores of the epoxy coating and improves the curing rate of the epoxy resin. Among the samples, SiC-G@GO_1_ and SiC-G@GO_2_ are excellent, being 4~5 orders of magnitude greater (Figure 12e), and the immersion time in 3.5 wt% NaCl aqueous solution is as long as 22 d, and the corrosion resistance is not significantly reduced, which means that SiC-G@GO_1_ and SiC-G@GO_2_ have long-term corrosion resistance. The order of the resistance arc radius of the modified coatings is SiC-G@GO_0.5_ < SiC-G@GO_1_ ≤ SiC-G@GO_2_, which means that, for SiC-G, the optimal addition amount of @GO is 1 wt% and 2 wt%, but 2 wt% is twice as much as 1 wt%, and the anti-corrosion performance is not significantly improved. The explanation for this phenomenon is that the addition of SiC-G@GO_0.1_ and SiC-G@GO_0.5_ is too small and cannot block the infiltration of the corrosive medium for a long time. Thus, it performs well in the early stage. SiC-G@GO_1_ and SiC-G@GO_2_ have an excellent corrosion resistance, which is attributed to the good dispersibility of SiC-G@GO_1_, while SiC-G@GO_2_ is agglomerated, but the increase in the total amount compensates for its agglomeration disadvantage. Therefore, compared with SiC-G@GO_1_, the performance of SiC-G@GO_2_ is not greatly improved. Due to the excessive amount of 2 wt%, the phenomenon of nanoparticle agglomeration occurs, so that the effect of the amount of 2 wt% cannot be fully exerted. Therefore, considering the action efficiency and cost, the addition amount of 1 wt% is the optimal addition ratio (the results obtained from the previous cross-sectional morphology observation and DSC curve analysis support this conclusion). 

The radius of the 2 wt% Nyquist plots began to increase on the 10th day, because the generation of corrosion products led to the emergence of a new resistance, which led to the increase in the overall resistance of the coating. The coating resistance increased further on the 15th and 21st days of immersion, which indicated that the corrosion products were increasing with the extension of the immersion time, which indicated that 2 wt% of the coating was immersed for a long time. In the case of SiC-G@GO, because of the defects in the coating caused by the cluster, it is not conducive to the long-term service of the coating. The radius of the 1 wt% Nyquist plots began to increase on the 22nd day, and the coating also increased slightly on the 22nd day, which indicates that less corrosion products were produced, and the resistance of the corrosion products increased less than that of the 2 wt% immersion time, indicating that the degradation rate of the coating and the generation rate of the defects were both lower than 2 wt%.

In the EP coatings containing 0.5 wt%, 1 wt%, and 2 wt% SiC-G@GO, the content of 0.5 wt% SiC-G@GO is too small to be effective in terms of the role played by SiC-G@GO. Table 3 shows the circuit diagrams for the fitting parameters and assignments based on the EIS data (Figure 4 and Figure 5). By analyzing the impedance spectrum, it can be concluded that the 10d are fitted to circuit diagram 4. This means that a corrosive medium reaches the metal substrate, but the degree of corrosion caused is relatively light. The reason for this is that the value of Z_f_ = 0.01 Hz does not drop much, indicating that only a small amount of corrosive medium penetrates the coating. When the filler is increased to 1 wt% and 2 wt%, 1 wt% only fits circuit diagram 2, which indicates that the corrosive medium cannot diffuse easily. This shows that the addition of 1 wt% SiC-G@GO fully exerts the excellent barrier properties of SiC-G@GO and significantly hinders the diffusion of the corrosive media. However, the 2 wt% SiC-G@GO was fitted to circuit diagram 3 after immersion for 21 d, because the added amount caused substantial agglomeration and was not enough to provide the full advantages of SiC-G@GO. Compared with the addition of 1 wt% SiC-G@GO, the barrier effect is somewhat reduced. Due to the strong impermeability of the 1 wt% application, the penetration of the corrosive medium is not enough to enable the instrument to detect that the coating is divided into three layers, thus fitting circuit diagram 2. The penetration resistance of 2 wt% is relatively weak. The penetration of the coating by the corrosive medium can be detected by the instrument as divided into three layers. Because the modulus of Z_f_ = 0.01 Hz almost fails to decrease, this indicates the fitting of circuit diagram 3 (Figure 4). The 0.5 wt% modulus at Z_f_ = 0.01 Hz decreases greatly, so that the circuit diagram 4 is fitted. According to the diffusion Formulas (1) and (2), the diffusion coefficient relationship between coatings with different addition amounts can be initially obtained: D_0_ < D_O.5_ < D_1_ > D_2_.

The reduction in porosity led to the coatings’ improved resistance to corrosion [48]. This is proved by the relationship between the micro-morphology and curing rate of the coating section and the EIS data. Because there are many micropores in pure EP, and the curing state is poor, the EIS data of the near-EP coating is poor. The EIS data of the coating correspond to the curing state and micro-morphology, respectively. This shows that a low porosity can improve the anti-corrosion performance of the coating. Therefore, it can be concluded that the anti-corrosion performance of the SiC-G@GO_1_ coating is optimal.

#### 3.5.2. Open Circuit Potential Analysis

The open circuit potential of EIS is closely related to the penetration of the corrosive medium. The more negative the potential and the greater the difference from the initial potential are, the more substantial the penetration of the corrosive medium will be [49]. The OCP values of pure epoxy and composite coatings were measured at different immersion times. According to Figure 13, it can be seen that the open circuit voltage of the coatings EP and SiC-G@GO_0.5_ decreases as a whole, indicating that with the prolongation of the soaking time, the corrosive medium gradually penetrates the coating, and the open circuit voltage of the coating decreases. The open circuit potential of 13 d dropped sharply, which indicated that the addition of 0.5 wt% pure epoxy was not enough to block the intrusion of the corrosive medium. However, the open circuit voltage of SiC-G@GO_1_ did not change greatly, indicating that it is difficult for the corrosive media to penetrate the coating. The general trend of SiC-G@GO_2_ with the prolongation of immersion time is the same as that of SiC-G@GO_1_, indicating that the content of ≥1 wt% SiC-G@GO has an excellent barrier effect and achieves a better corrosion protection performance.

#### 3.5.3. Potentiodynamic Polarization Measurements

The corrosion resistance of the filler was studied using a potentiodynamic polarization test. The polarization curve of the sample after immersion in 3.5 wt% NaCl aqueous solution is shown in Figure 8. The electrochemical parameters include the corrosion current density (*i_corr_*), corrosion potential (*E_corr_*), and anodic (*βa*) and cathodic (*βc*) Tafel slopes. Because different fillers have great differences in performance with respect to the improvement of epoxy coatings, different immersion times are used for the measurements, as shown in Table 4. The longer the immersion time is, the more significant the penetration of the corrosive medium will be, and the worse the measured data at this time will be. The corrosion current density (*i_corr_*) is obtained from the Tafel plot by extrapolating the straight line portion of the corrosion potential curve (*E_corr_*) [49].

As shown in Figure 14a, the bare iron substrate was immersed in the solution for the shortest time, and the obtained corrosion current was the largest. The corrosion current of EP, GO/EP, and SiC/EP after immersion for 10 d is one order of magnitude smaller than that of the bare immersion for 10 min (Figure 14b–d), which indicates that the epoxy coating has a protective effect on the iron substrate. For the epoxy coatings with SiC-G@GO fillers (Figure 14e–g), the E_corr_ values showed more positive potentials, and the change in the anodic Tafel slope was larger than that of the cathodic slope (Table 4). The larger the slope is, the larger the reaction resistance is, the smaller the corrosion rate is, and the easier it is for corrosion to occur. After immersion for 25 d, the SiC-G@GO_1_ sample had the smallest corrosion current density (*I_corr_*), indicating that the SiC-G@GO_1_ sample has the best protection performance, while compared with SiC-G@GO_2_, SiC-G@GO_1_ shows a smaller change in the anode Tafel slope than the cathode slope. This shows that the corrosion resistance of SiC-G@GO_2_ is lower than that of SiC-G@GO_1_ after prolonged immersion. In this experimental analysis, the corrosion rate (CR, mm year^−1^) and the increase in the corrosion efficiency (EPE, %) of the samples were used to quantitatively evaluate the protective performance of each coating. CR represents the uniform corrosion rate of the Fe matrix in the coating sample, and CPE% reflects the improvement of the protective performance of the coating after adding the fillers. The CR is calculated according to the following formula:(9)$$\mathrm{CR}=\frac{kM_{m}I_{corr}}{n\rho_{m}}\;$$

Above, *k* is a constant, 3268.5 mol A^−1^; *M_m_* is the molar mass of the metal, g mol^−1^; *n* is the charge transfer tree of the metal corrosion reaction; and *ρ_m_* is the density of the metal, g cm^−3^. The EPE% is calculated using the following formula:(10)$$\mathrm{EPE}\%=\frac{I_{corr,I}-I_{corr,i}}{I_{corr,I}}\times 100\%\;$$

Above, *I_corr_,_I_* is the corrosion current density of EP, A cm^−2^, and *I_corr_,_i_* is the corrosion current density of the coating sample *i*, A cm^−2^. The calculation results of the CR and EPE% are shown in Table 4. It can be seen from the table that the value of the CR of the bare iron base sample is 0.268 mm year^−1^. The values for the EP, GO_1_, and SiC_1_ samples are 0.011, 0.014, and 0.0066 mm year^−1^, respectively. This shows that the SiC_1_ filler can improve the protective performance of EP, whereas the GO_1_ filler can not only not increase the long-term protection but can also reduce the protective performance of EP. This result of the polarization test contradicts the theoretical analysis of the maze effect. However, for conductive fillers, we believe that this presence is justified. Because the introduction of conductive fillers into the coating may increase the conductivity of the coating, a non-corrosive polarization current may be detected on the surface of the conductive coating during the polarization test, affecting the test results. Therefore, the polarization curve of GO cannot faithfully reflect the corrosion state of the metal matrix. The addition amount of the modified composite filler SiC-G@GO, from small to large, is 1.94, 5.38, 1.17 × 10^−3^, and 0.127 μm year^−1^. The results show that the SiC-G@GO_1_ filler can significantly improve the protective performance of EP.

### 3.6. Analysis of the Neutral Salt Spray Test

In order to visually verify the anticorrosion performance of the composite, Figure 15 shows the macrophotograph of the coating after its exposure to a salt spray environment for 20 d. The conditions of the Neat EP, GO/EP, and SiC/EP coatings are shown in Figure 15a–c, indicating the presence of pitting, coating peeling, and corrosion product build-up, which can be attributed to the electrolyte passing under the coating through the scribe line and the propagation of the micropores present in the coating structure [50]. In particular, the SiC/EP coating exhibits a poor adhesion, which is due to the severe agglomeration of the nanoparticles that affects the adhesion of the epoxy coating to the substrate. SiC-G@GO_0.5_ has more corrosion in places without scratches, because an addition that is too minimal is not enough to block the corrosion medium. SiC-G@GO_2_ has substantial corrosion product accumulation at the defects because excessive amounts of SiC-G@GO affect the adhesion of the coating, and the corrosive medium easily penetrates from the site of the scratch, causing corrosion. The addition of 1 wt% can not only block the corrosion medium, but also has little impact on the adhesion, so SiC-G@GO_1_ The accumulation of corrosion products is relatively small. It is concluded that the corrosion resistance of the SiC-G@GO_1_ coating is the best.

### 3.7. Schematic Diagram of the Mechanism

The corrosion protection mechanisms of the various coating systems are shown in Figure 16. The exposed iron-based surface undergoes a corrosion reaction in humid air, as shown in Figure 16a, consuming oxygen in the air to generate hydrogen and Fe_3_O_2_. For the pure epoxy coatings (Figure 16b), the corrosive electrolyte easily passes through the solvent-based micropores created by solvent evaporation during the epoxy coating’s curing and penetrates the coating. After adding SiC and GO (Figure 16c,d), the anticorrosion performance of the composite coating was improved compared with the pure epoxy coating. The improved corrosion resistance is attributed to the increased number of diffusion paths for the corrosive electrolyte caused by SiC and GO. Unfortunately, the accumulation of the agglomerates of SiC and GO in the epoxy resins leads to the penetration of the coating–metal substrate interface by corrosive electrolytes through defects in the coating and new pathways. However, the addition of the SiC-G@GO composite filler resulted in a significant increase in the corrosion resistance of the composite coating compared to SiC/EP and GO/EP (Figure 14d). The protection mechanism of SiC-G@GO/EP is as follows (Figure 16e). The functionalized SiC nanoparticles act as spacers between the GO sheets, which enables the SiC-G@GO composite filler to exhibit an excellent dispersibility and stability in the epoxy matrix, and through the further reaction of functional groups, the composite filler is improved with respect to the interfacial adhesion between the filler and epoxy matrix. Therefore, the highly dispersed SiC-G@GO composite filler can reduce more defects and micropores in the epoxy coating and reduce the permeability of the coating, but excessive SiC-G@GO reduces the adhesion of the epoxy coating (Figure 16f). In addition, the impermeability of the GO nanosheets significantly improves the barrier properties of epoxy coatings and increases the diffusion of corrosive electrolytes into the coating and substrate through tortuous diffusion paths. More importantly, the addition of GLYMO can not only improve the dispersion of SiC and GO in the epoxy resin through surface functionalization but also increase the number epoxy resin crosslinking sites through the epoxy groups, making the crosslinking density denser and thereby improving the barrier.

From the above analysis, we can draw the conclusion that the anti-corrosion performance of a filler exhibits an extreme value when the concentration is 0.5~2 wt%. In particular, in the salt spray experiment, it was found that there were more corrosion products around the scratches of the coating with a content of 2 wt%. Therefore, it can be judged that the cathodic stripping is relatively significant [51], which limits the concentration in the subsequent experiments.

### 3.8. Tribological Properties of the Composite Coatings

The friction coefficients and wear rates of the four coatings are shown in Figure 17. Figure 17a shows the dynamic curves of the friction coefficients of all five coatings. Clearly, due to the presence of some micro-protrusions on the composite surface, all the coating samples exhibited a brief ‘saturation’ phase at the beginning of the wear process (phase I, marked in Figure 17a), and during this period, the friction coefficient increased with the speed, decreasing with the increasing amount of SiC-G@GO [52]. As the wear process continued, the friction coefficients of most of the composite coatings were basically in a relatively stable state, without significant changes. The difference is that the friction coefficient of the pure EP coating always increases at this stage because the pure epoxy coating itself has micropores. These micropores can damage the structure of the epoxy coating when subjected to friction with a large normal load, so that the coefficient of the friction increases relatively rapidly. When the filler SiC-G@GO ≥ 0.5 wt% is applied, GO fills the micropores of the epoxy coating to a certain extent, thereby enhancing the mechanical properties of the overall structure of the coating. Additionally, the surface of the SiC-G@GO filler contains (-OH) and epoxy groups that function to improve the curing rate of the epoxy resin, which provides the epoxy coating with a higher crosslinking density. The dynamic curve of the friction coefficient also illustrates that the higher the filler content is, the lower the friction coefficient in the plateau phase of the second stage will be. Finally, the friction coefficient remained stable (phase III, marked in Figure 17a), and during this phase, the friction coefficient experienced a period of decline, a phenomenon that may be due to a greater amount of GO acting as a solid lubricant [53]. During the friction process, a lubricating friction film is gradually formed. When a stable lubricating friction film is formed, the friction coefficient is gradually stabilized. It can be seen that the friction coefficients of SiC-G@GO_1_ and SiC-G@GO_2_ tended to remain consistent, indicating that a greater amount of SiC-G@GO does not improve the long-term wear resistance of epoxy coatings.

Figure 17b presents the average coefficients of friction and wear rates of the four coatings. Obviously, the average friction coefficient of the pure EP coating is the largest, being 0.807, while the coatings with the fillers decrease sequentially: SiC-G@GO_0.5_ > SiC-G@GO_1_ > SiC-G@GO_2_. SiC-G@GO_2_ reduces the friction coefficient of the EP coating to 0.331, and the average friction coefficient of the SiC-G@GO_1_ coating is 0.472. In addition, the wear resistance of the coatings is reflected by the wear rate (WR:(3),(4)), and the wear rates of the SiC-G@GO_1_ and SiC-G@GO_2_ composite coatings are 2.66 × 10^−5^ and 2.23 × 10^−5^ g•N^−1^•m^−1^, which are about 56.6% and 63.5% lower than that of the pure EP coating (6.13 × 10^−5^ g•N^−1^•m^−1^), respectively. SiC-G@GO_2_ is better than SiC-G@GO_1_ in terms of the average friction coefficient and wear rate, which is due to the fact that a greater amount of SiC-G@GO performs better in the early stage of friction.

Figure 18a–d shows the 3D morphology and corresponding cross-sectional width and depth of the wear traces for the four coatings. Macroscopically, the overall appearance of the wear marks is somewhat similar, but the degree of damage and amount of wear are quite different. Clearly, the blank EP coating without SiC-G@GO shows the largest wear width and depth (Figure 18a). Under the action of SiC-G@GO, the wear damage of the composite coating decreases with the increase in the filler content, and the wear trace becomes narrower and shallower. For the SiC-G@GO_1_ and SiC-G@GO_2_ composite coatings, the depth reduction is more pronounced (Figure 18c,d). Because a greater amount of SiC-G@GO performs better in the early stage of friction, the wear trace of SiC-G@GO_2_ over a short time is narrower and shallower, but the kinetic friction coefficients of SiC-G@GO_1_ and SiC-G@GO_2_ gradually increase. Consistently, in long-term applications, 1 wt% SiC-G@GO can obtain a significant performance improvement with less addition. Thus, 1 wt% is the optimal addition ratio.

### 3.9. Research on the Mechanism of the Wear Resistance

The wear damage and wear mechanism of the composite coating were further studied by SEM and EDS based on the morphology and micro-area element analysis of the SiC-G@GO_1_ wear scar, as shown in Figure 19. Combined with the element content and distribution of the local area (Figure 19b), the location of the SiC-G@GO filler can be roughly judged, and it can be seen that the distributions of the SiC elements and O elements are roughly the same (Figure 19d,e). This is because of the grafting of SiC and GO. From this area, it can be seen that the SiC elements diffuse from the point of concentration to the surrounding area (Figure 19e), and SiC acts as a similar rolling friction when the GO surface is rolled to the surrounding area. This shows that, in the whole friction process, the lamellar GO and granular SiC form an interface similar to rolling friction during frictional movement, which greatly reduces the damage of the friction body when the epoxy coating is directly rubbed. Thus, the original friction on the fragile epoxy coating was transformed into the rolling friction performed by SiC on GO, with an excellent mechanical strength (Figure 19f). SiC-G@GO has an excellent load-carrying capacity and wear resistance, and the improved wear resistance of the epoxy coatings can be attributed to the composite fillers’ improvement of the curing state of the epoxy coatings, the formation of the tribofilm and microstructure, and other comprehensive factors.

## 4. Conclusions

Taking rail transit vehicles as an example, the dynamic service equipment is inevitably affected by wear, fatigue, corrosion, and other types of damage due to the interaction between multiple environments. Thus, research on the functional coatings used for corrosion and wear resistance and their development has been carried out.

In conclusion, compared with SiC and GO, the incorporation of SiC-G@GO composite fillers into epoxy coatings significantly improves their wear and corrosion resistance. According to the electrochemical measurements and salt spray tests, the enhanced anticorrosion performance is attributed to the uniformly distributed nature of the SiC-G@GO composites. The 1 wt% SiC-G@GO is the optimal addition ratio, enabling the epoxy composite coating with the best corrosion resistance to be prepared, which increases the modulus value of the coating at Z_f_ = 0.01 Hz by 3~4 orders of magnitude. At the same time, SiC and GO interact with each other. SiC and GO synergistically enhance the wear resistance of the epoxy resin. Under a 30 N load, the average friction coefficient of the composite coating with a 1 wt% content is 41.5% lower than that of the pure EP coating, and the wear rate is 56.6% lower. This provides a positive reference value for the application of SiC and GO in the field of wear and anti-corrosion research.

## Figures and Tables

**Figure 1 polymers-14-04704-f001:**
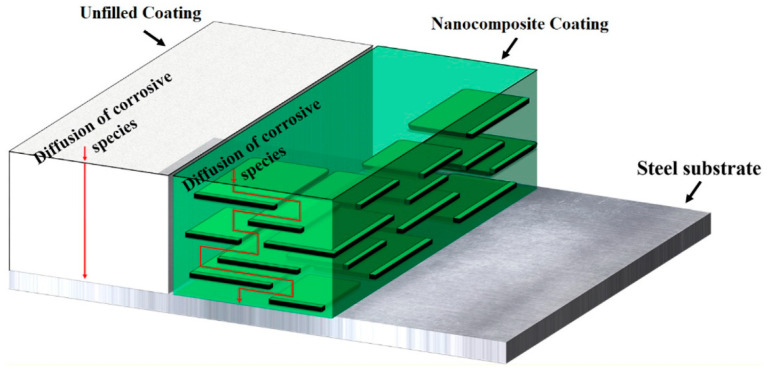
The labyrinth effect of 2D nanolayers on the barrier properties of an organic coating [32].

**Figure 2 polymers-14-04704-f002:**
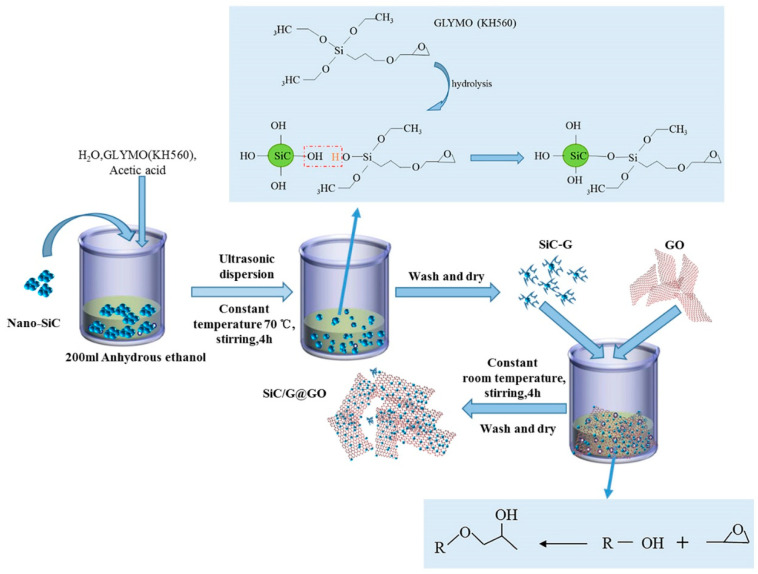
Schematic diagram of the preparation of the SiC-G@GO composite filler.

**Figure 3 polymers-14-04704-f003:**
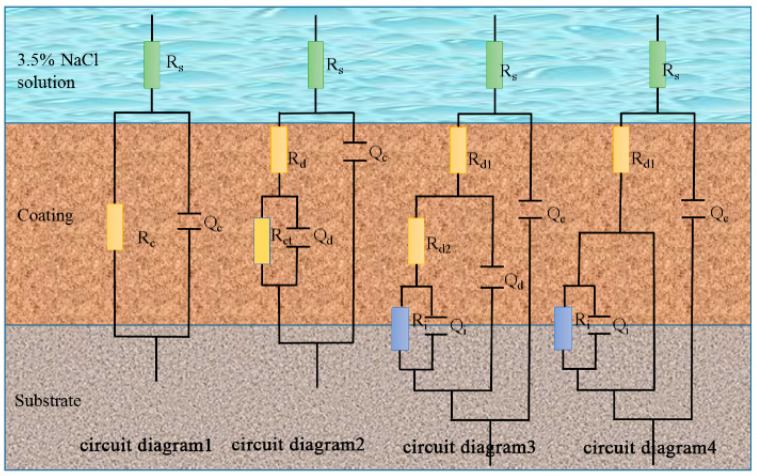
The corresponding equivalent circuit.

**Figure 4 polymers-14-04704-f004:**
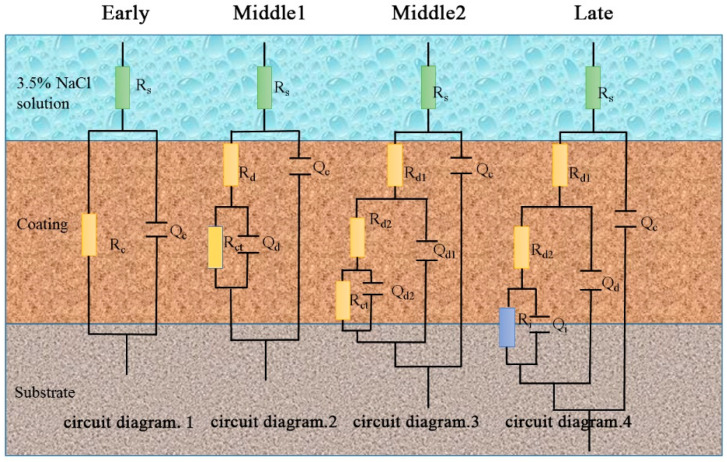
Equivalent circuit diagrams of SiC-G@GO nanocomposite coatings with different ratios.

**Figure 5 polymers-14-04704-f005:**
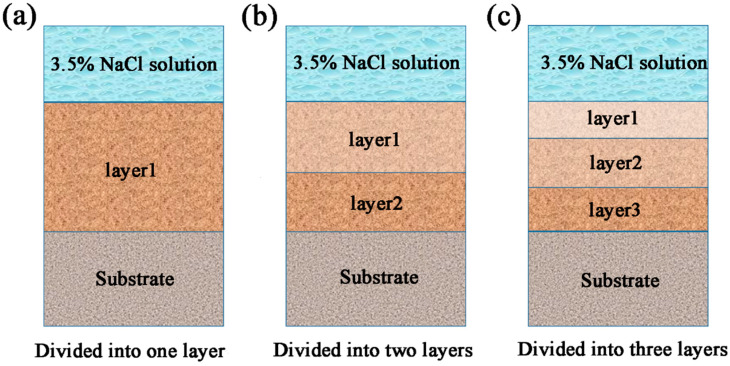
Schematic diagram of layering. (**a**) When the coating is just soaked, the corrosive medium does not invade, and it is divided into one layer. (**b**) The corrosive medium intrusion is divided into two layers. (**c**) The corrosive medium intrusion is divided into three layers.

**Figure 6 polymers-14-04704-f006:**
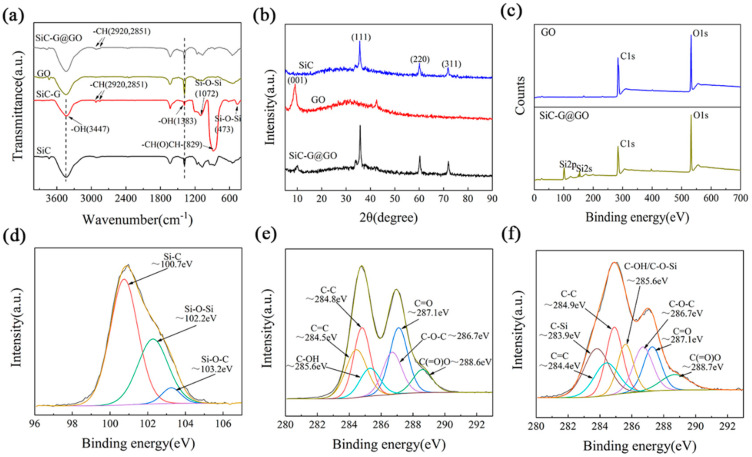
(**a**) FT-IR patterns of the SiC, GO, SiC-G, and SiC-G@GO samples; (**b**) XRD patterns of thw SiC, GO, and SiC-G@GO samples; (**c**–**f**) XPS spectra and fitting results of the GO and SiC-G@GO samples; (**c**) XPS full–scan spectra of the GO and SiC-G@GO composites; (**d**) SiC-G@GO SiC2p high-resolution spectrum; (**e**) SiC-G C1s high-resolution spectrum; and (**f**) SiC-G@GO composite C1s high-resolution spectrum.

**Figure 7 polymers-14-04704-f007:**
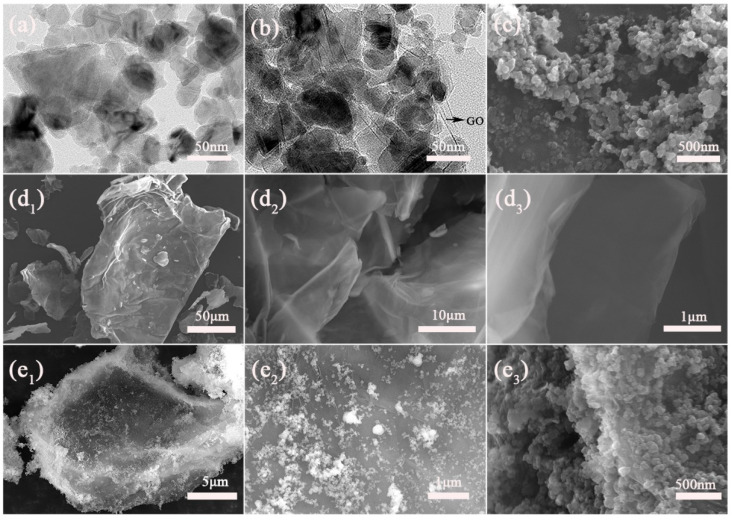
(**a**,**b**) TEM images of the (**a**) SiC and (**b**) SiC-G@GO samples, the black arrow is GO.; (**c**–**e**) FE-SEM images of (**c**) SiC, (**d**) GO, pictures with different magnification(**d_1_**–**d_3_**), and (**e**) SiC-G@GO, pictures with different magnification (**e_1_**–**e_3_**).

**Figure 8 polymers-14-04704-f008:**
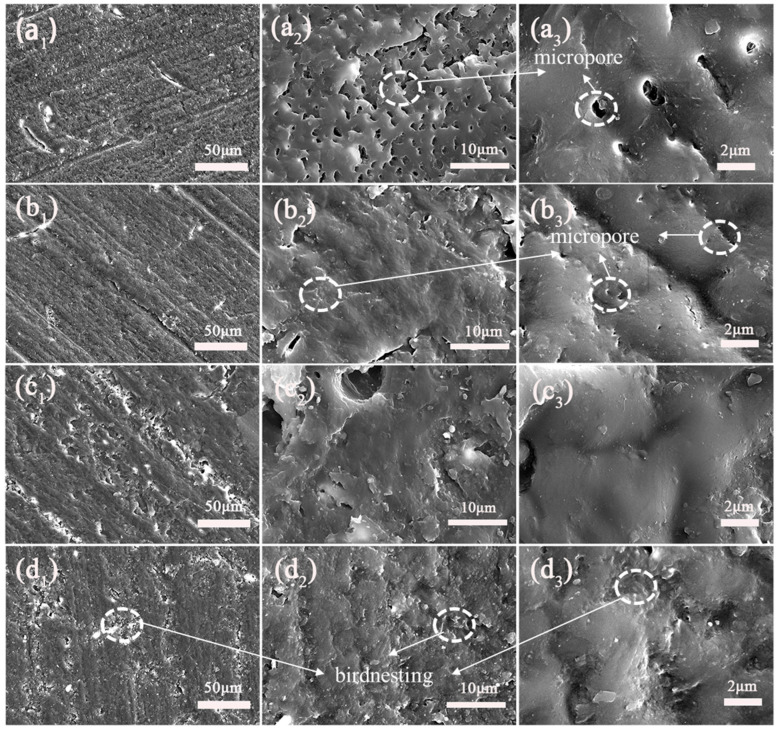
FE-SEM images of coating fracture surface at different magnification: (**a**) neat EP, pictures, (**b**) SiC-G@GO_0.5_, (**c**) SiC-G@GO_1_, and (**d**) SiC-G@GO_2_ coatings.

**Figure 9 polymers-14-04704-f009:**
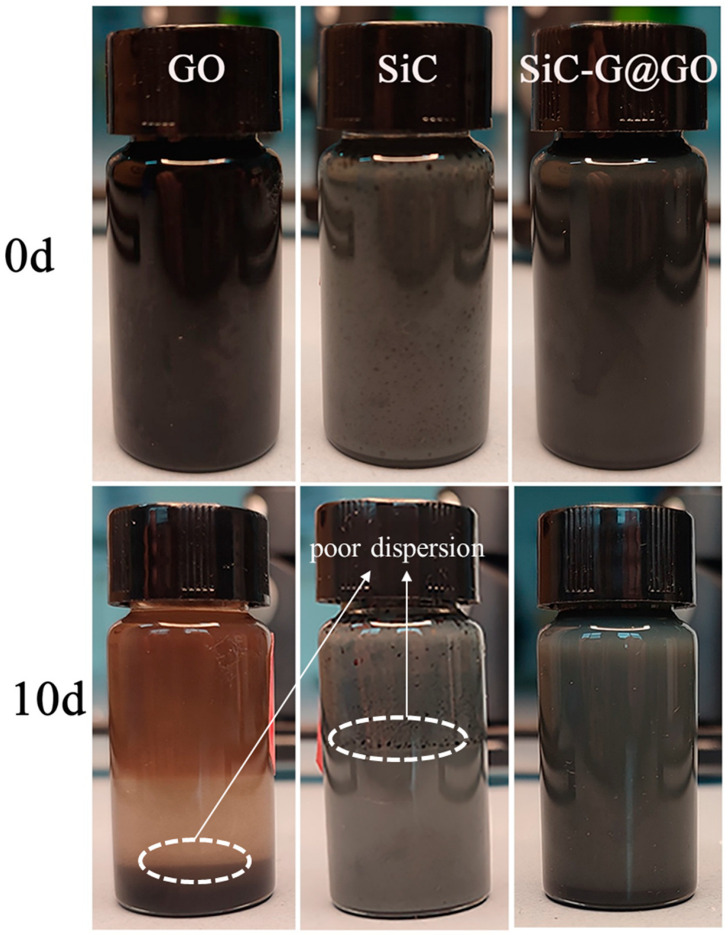
SiC, GO, and SiC-G@GO: dispersion stability in EP.

**Figure 10 polymers-14-04704-f010:**
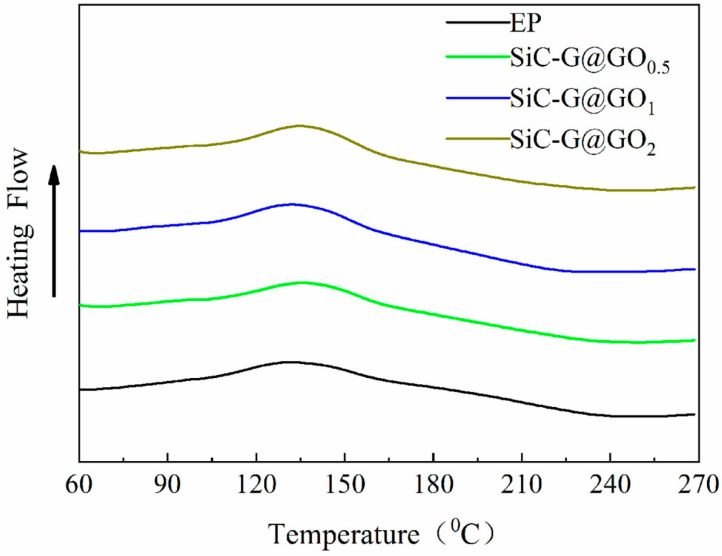
DSC thermograms of the neat EP and EP-based composite coatings obtained at a heating rate of 10 °C/min.

**Figure 11 polymers-14-04704-f011:**
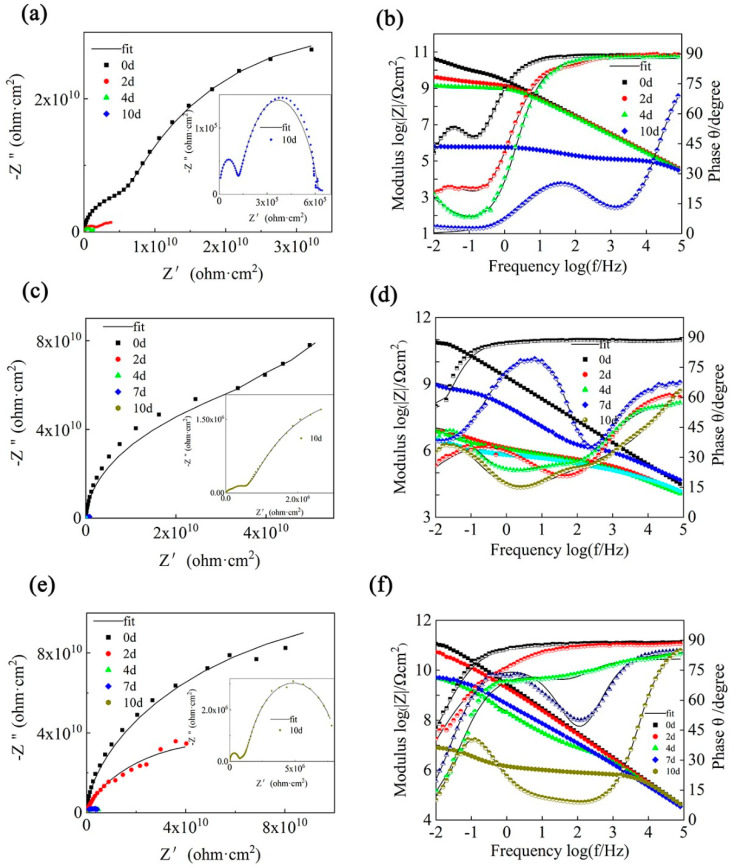
(**a**,**c**,**e**) Bode and (**b**,**d**,**f**) Nyquist diagrams; (**a**,**b**) the EP coating, (**c**,**d**) GO_1_ coating, and (**e**,**f**) SiC_1_ coating at different immersion times in 3.5 wt% NaCl solution.

**Figure 12 polymers-14-04704-f012:**
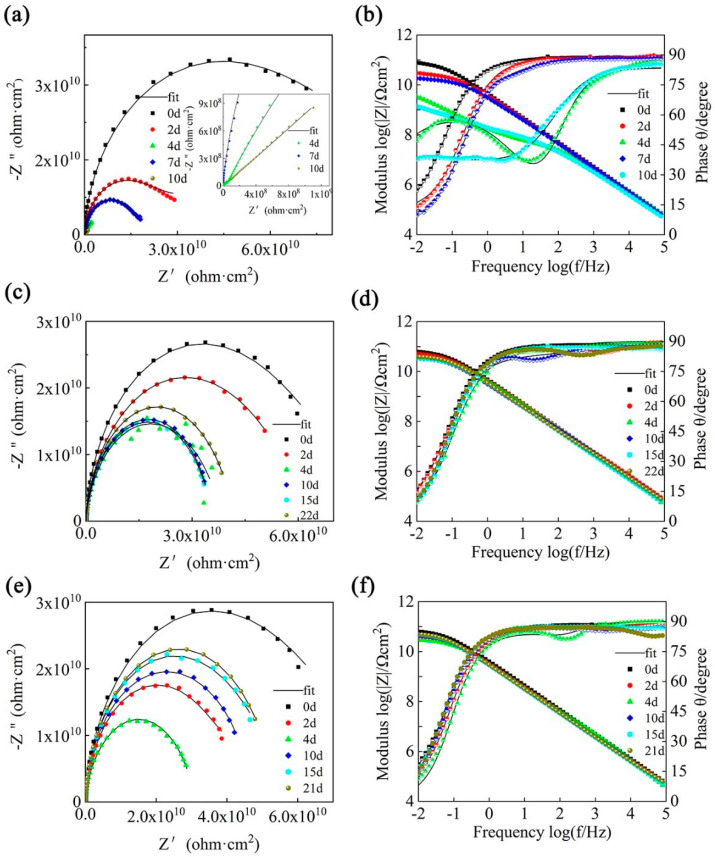
(**a**,**c**,**e**) Bode and (**b**,**d**,**f**) Nyquist diagrams of the (amb) SiC-G@GO_0.5_ coating, (**c**,**d**) SiC-G@GO_1_ coating, and (**e**,**f**) SiC-G@GO_2_ coating at different immersion times in 3.5 wt% NaCl solution.

**Figure 13 polymers-14-04704-f013:**
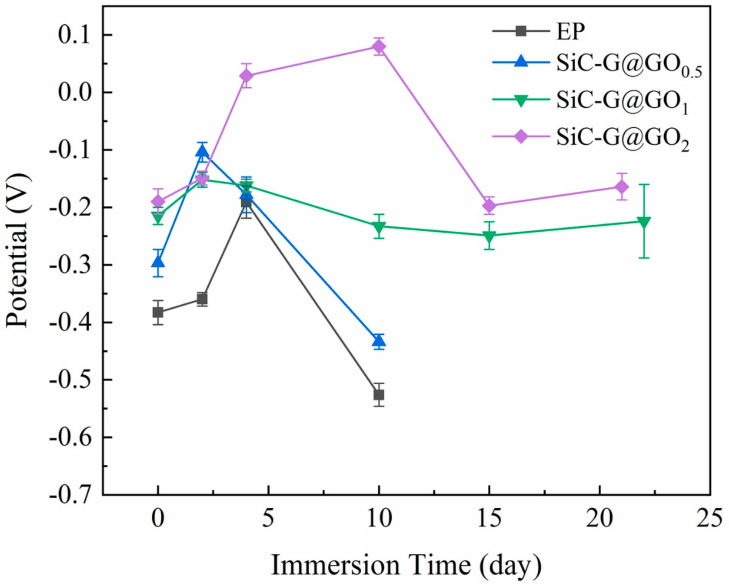
The variation in the OCP values of different coatings as a function of the immersion time (the scatter band reveals the data variation range over the average value of three replicates).

**Figure 14 polymers-14-04704-f014:**
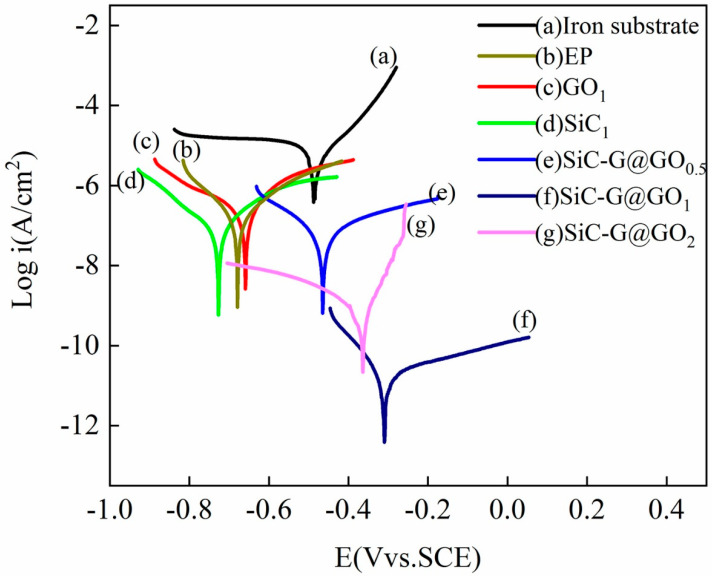
Polarization curves of the coated samples.

**Figure 15 polymers-14-04704-f015:**
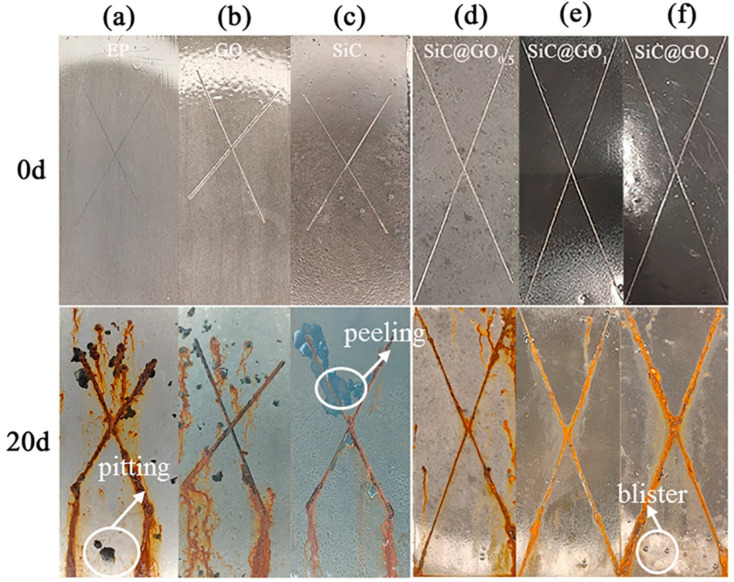
Images of epoxy coatings containing (**a**) neat EP, (**b**) GO_1_, (**c**) SiC_1_, (**d**) SiC-G@GO_0.5_, (**e**) SiC-G@GO_1_, and (**f**) SiC-G@GO_2_ before and after exposure to the salt spray test for 20 d.

**Figure 16 polymers-14-04704-f016:**
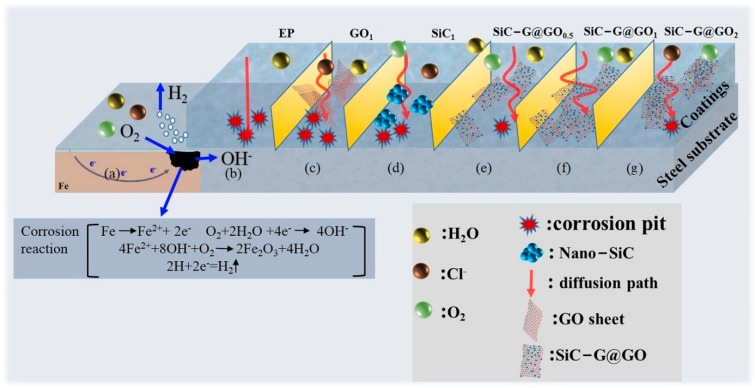
(**a**) Corrosion of iron-based surfaces exposed to air. Anticorrosion mechanism of (**b**) neat EP, (**c**) GO_1_, (**d**) SiC_1_, (**e**) SiC-G@GO_0.5_, (**f**) SiC-G@GO_1_, and (**g**) SiC-G@GO_2_.

**Figure 17 polymers-14-04704-f017:**
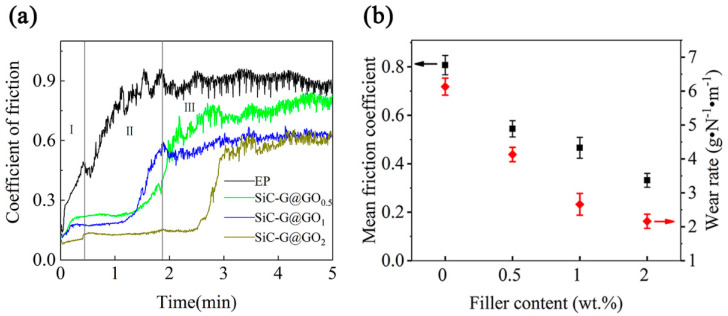
Tribological properties of the experimental coatings: (**a**) kinetic friction coefficient; (**b**) average friction coefficient and wear rate.

**Figure 18 polymers-14-04704-f018:**
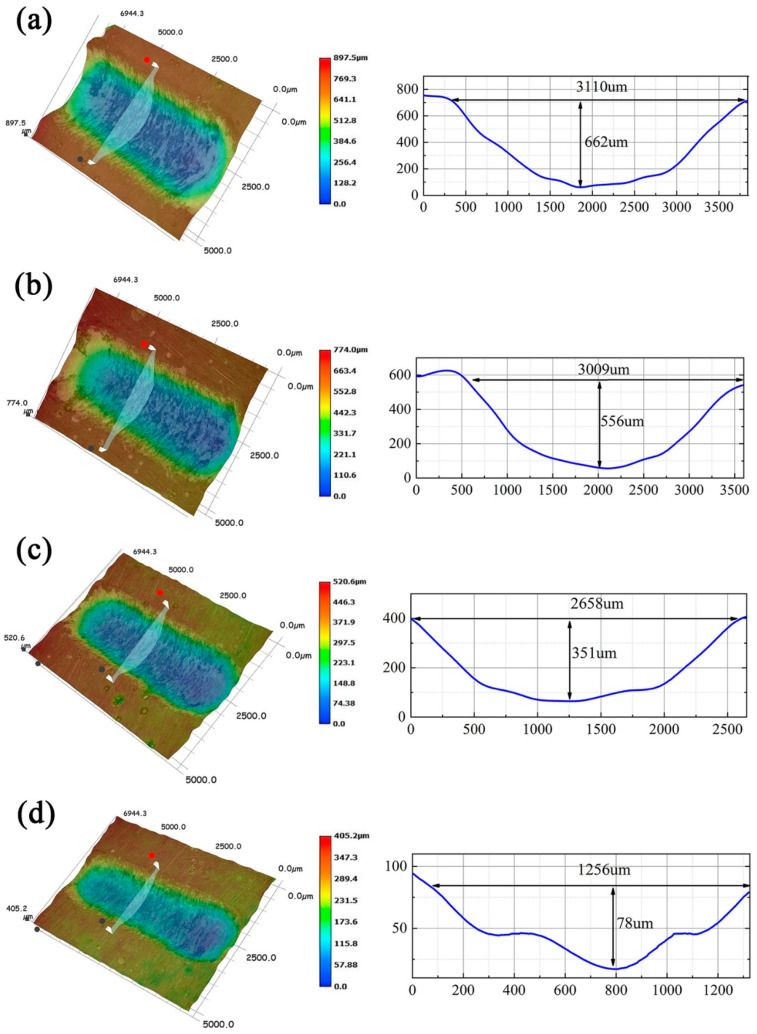
(**a**–**d**) The 3D morphologies and corresponding wear scar cross-sectional profiles of the four coatings: (**a**) EP coating, (**b**) SiC-G@GO_0.5_ coating, (**c**) SiC-G@GO_1_ coating, and (**d**) SiC-G@GO_2_ coating.

**Figure 19 polymers-14-04704-f019:**
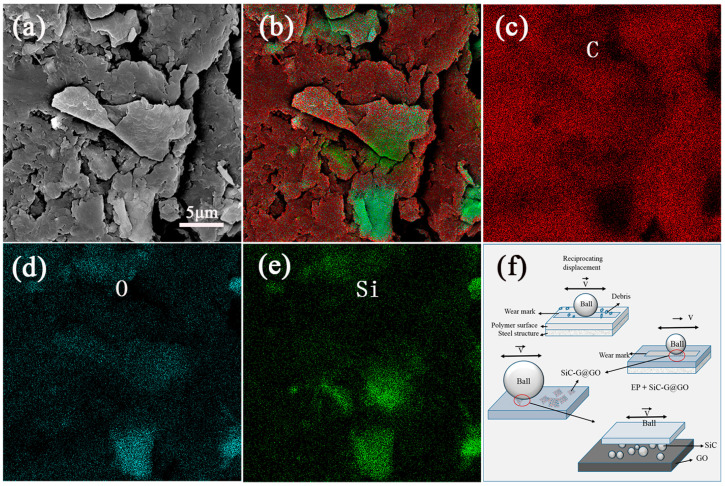
(**a**) FE-SEM image of SiC-G@GO_1_ wear scars; (**b**–**e**) EDS images of C, O, and Si elements of SiC-G@GO_1_ wear scars; (**f**) schematic diagram of rolling friction.

**Table 1 polymers-14-04704-t001:** Cure characteristics of the neat EP and EP-based composite coatings.

Samples	β (°C/min)	*ΔT* (°C)	*ΔH*	*ΔT**	*ΔH**	*CI*
** *EP* **	10	61.66	0.043	1.00	1.00	1.00
** * SiC-G@GO_0.5_ * **	10	59.03	0.049	0.95	1.14	1.08
** *SiC-G@GO_1_* **	10	51.28	0.052	0.83	1.21	1.01
** *SiC-G@GO_2_* **	10	59.02	0.049	0.95	1.14	1.08

**Table 2 polymers-14-04704-t002:** EIS data of the EP, GO, and SiC nanocomposite coatings.

Samples	Time	CPE_1_		R_1_(ohmcm^2^)	CPE_2_		R_2_(ohmcm^2^)	CPE_3_		R_3_(ohmcm^2^)	Circuit Diagram
		Y_0_(ohm^−1^cm^−2^s^n^)	n		Y_0_(ohm^−1^cm^−2^s^n^)	n		Y_0_(ohm^−1^cm^−2^s^n^)	n		
**EP**	0 d	5.391 × 10^−11^	0.989	9.364 × 10^9^	1.965 × 10^−10^	0.881	6.387 × 10^10^	-	-	-	2
	2 d	6.452 × 10^−11^	0.976	1.295 × 10^9^	1.073 × 10^−9^	0.471	6.573 × 10^8^	-	-	-	2
	4 d	6.83 × 10^−11^	0.982	5.718 × 10^8^	9.533 × 10^−11^	0.671	6.158 × 10^8^	-	-	-	2
	10 d	1.634 × 10^−10^	0.8	1.379 × 10^7^	1.632 × 10^−7^	0.8	5.637 × 10^5^	-	-	-	4
**GO_1_**	0 d	8.437 × 10^−11^	0.976	1.347 × 10^11^	-	-	-				1
	2 d	6.928 × 10^−9^	0.708	4.286 × 10^7^	3.3 × 10^−7^	0.513	1.601 × 10^7^	-	-	-	2
	4 d	1.36 × 10^−8^	0.669	3.366 × 10^5^	3.463 × 10^−7^	0.427	3.436 × 10^6^	4.238 × 10^−7^	1	5.294 × 10^7^	3
	7 d	9.904 × 10^−10^	0.763	2.179 × 10^7^	1.046 × 10^−9^	0.999	5.043 × 10^8^		-	-	4
	10 d	3.818 × 10^−8^	0.563	4.133 × 10^5^	1.297 × 10^−6^	0.450	3.829 × 10^8^	-	-	-	4
**SiC_1_**	0 d	5.734 × 10^−10^	0.977	3.109 × 10^11^	-	-	-	-	-	-	1
	2 d	6.867 × 10^−11^	0.974	1.731 × 10^10^	5.024 × 10^−11^	0.565	1.195 × 10^11^	-	-	-	2
	4 d	1.302 × 10^−10^	0.928	8.872 × 10^6^	7.839 × 10^−10^	0.853	4.989 × 10^9^	-	-	-	2
	7 d	2.246 × 10^−10^	0.895	4.454 × 10^7^	2.519 × 10^−10^	0.756	5.754 × 10^9^	-	-	-	2
	10 d	1.05 × 10^−10^	0.934	7.931 × 10^5^	3.501 × 10^−7^	0.615	1.16 × 10^7^	-	-	-	4

**Table 3 polymers-14-04704-t003:** EIS data of the SiC-G@GO nanocomposite coatings with different ratios.

Samples	Time	CPE_1_		R_1_(ohmcm^2^)	CPE_2_		R_2_(ohmcm^2^)	CPE_3_		R_3_(ohmcm^2^)	Circuit diagram
		Y0(ohm^−1^cm^−2^s^n^)	n		Y0(ohm^−1^cm^−2^s^n^)	n		Y0(ohm^−1^cm^−2^s^n^)	n		
**SiC-G@GO_0.5_**	0 d	3.716 × 10^−11^	0.973	8.643 × 10^11^	-	-	-	-	-	-	1
	2 d	2.582 × 10^−11^	0.335	1000	3.727 × 10^−11^	0.982	4.153 × 10^10^	-	-	-	2
	4 d	6.835 × 10^−11^	0.927	4.036 × 10^7^	1.712 × 10^−9^	0.667	1.691 × 10^11^	-	-	-	2
	7 d	4.015 × 10^−11^	0.8	6.55 × 10^8^	3.097 × 10^−11^	0.8	1.905 × 10^10^				3
	10 d	7.97 × 10^−11^	0.938	2.777 × 10^7^	9.176 × 10^−11^	0.884	4.002 × 10^7^	2.513 × 10^−7^	0.473	1.365 × 10^10^	4
**SiC-G@GO_1_**	0 d	3.528 × 10^−11^	0.969	6.776 × 10^10^	-	-	-	-	-	-	1
	2 d	4.186 × 10^−10^	0.946	4.786 × 10^10^	-	-	-	-	-	-	1
	4 d	3.945 × 10^−10^	0.970	3.282 × 10^10^	-	-	-	-	-	-	1
	10 d	3.819 × 10^−11^	0.8	6.779 × 10^9^	2.123 × 10^−11^	0.8	3.084 × 10^10^	-	-	-	2
	15 d	4.793 × 10^−11^	0.965	2.128 × 10^10^	6.974 × 10^−11^	0.699	1.452 × 10^10^	-	-	-	2
	22 d	4.996 × 10^−11^	0.947	2.669 × 10^9^	9.706 × 10^−12^	0.8	5.363 × 10^10^	-	-	-	2
**SiC-G@GO_2_**	0 d	4.114 × 10^−11^	0.970	7.564 × 10^11^	-	-	-	-	-	-	1
	2 d	5.398 × 10^−11^	0.958	4.014 × 10^10^	-	-	-	-	-	-	1
	4 d	5.457 × 10^−10^	0.955	2.765 × 10^10^	-	-	-	-	-	-	1
	10 d	5.924 × 10^−11^	0.959	3.626 × 10^10^	2.539 × 10^−10^	0.788	1.093 × 10^10^	-	-	-	2
	15 d	5.684 × 10^−11^	0.969	4.146 × 10^10^	1.405 × 10^−10^	0.829	2.11 × 10^10^	-	-	-	2
	21 d	4.018 × 10^−11^	0.978	4.956 × 10^5^	1.652 × 10^−11^	0.929	3.529 × 10^10^	7.325 × 10^−11^	0.695	1.861 × 10^10^	3

**Table 4 polymers-14-04704-t004:** The electrochemical corrosion parameters derived from the potentiodynamic polarization test.

Samples	Soaking Time	*E_corr_* (Vvs.SCE)	*i_corr_* (A/cm^2^)	*β a* (V/dec)	−*β c* (V/dec)	CR (cm Year^−1^)	EPE%	Coating Thickness (μm)
** *Iron substrate* **	10 min	−0.487	3.46 × 10^−6^	0.104	−0.439	2.68 × 10^−2^	-	0
** *EP* **	10 d	−0.674	1.412 × 10^−7^	0.125	0.112	1.1 × 10^−3^	-	101.3
** *GO_1_* **	10 d	−0.655	1.862 × 10^−7^	0.095	0.199	1.4 × 10^−3^	−31	98.2
** *SiC_1_* **	10 d	−0.659	8.884 × 10^−8^	0.142	0.276	6.65 × 10^−4^	37	103.5
** *SiC-G@GO_0.5_* **	10 d	−0.466	6.95 × 10^−8^	0.259	0.167	5.38 × 10^−4^	50	97.3
** *SiC-G@GO_1_* **	25 d	−0.311	1.51 × 10^−11^	0.268	0.081	1.17 × 10^−7^	99	96.6
** *SiC-G@GO_2_* **	25 d	−0.404	1.64 × 10^−9^	0.037	0.251	1.27 × 10^−5^	98	99.7

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
