# Peer review of "A Micro-Nano Structure Formed by SiC/Graphene Oxide Self-Assembly Improves the Wear Resistance and Corrosion Resistance of an Epoxy-Based Composite Coating"

_polymers, 2022, doi:10.3390/polym14214704_

Round 1
Reviewer 1 Report
Please see attached for my detailed comments. Good luck!

Reviewer 2 Report
Comments on polymers-1989187
The manuscript entitled “Micro nano structure formed by SiC/graphene oxide self-asssembly improves wear resistance and corrosion resistance of epoxy-based composite coating” presents an epoxy-based composite coating through the self-assembly of Silicon carbide (SiC) and graphene oxide (GO) and tuning the interfacial structure with epoxy resin. The coatings were comprehensively characterized, including electrochemical behavior, salt spray test, and friction and wear experiments.
The manuscript is well-written, results are well-discussed, however, there are several amendments required to be resolved before accepting it for publication which are disclosed below:
· The authors should provide the full forms in the very starting, for example, what is EP in the Abstract.
· Several sentences are very long, please try to make them precise and to the point. For example, the very first sentence of the abstract, and the very first sentences of the Introduction. Please have a look carefully.
· The authors should provide relevant references for the sentence “but also reduce the adhesion of the coating.”
· Please recheck the second last paragraph of the introduction, the first sentence has a few words which are repetitive and read a bit confusing.
· The authors should explain what is labyrinth effect in the Introduction with proper references.
· Thorough references should be cited in the Introduction, which is missing.
· Figure 1 is not very clear, the font of the labeling should be increased for better understanding. And the chemical structure should be more visible.
· In section 2.3.2, please check the line “and the third layer that is not immersed. for three layers.” There are several mistakes like that in the submitted manuscript.
· The format of Figure 5 is not uniform and the resolution is also very poor. Please increase the resolution and provide more visible figures.
· The authors are highly suggested to show SEM images with different magnifications for fillers as well as for coatings; moreover, it is better to label the images for better understanding.
· How do the authors assure the dispersion of particles in the matrix ?? The authors should provide some experimental results to assure the homogenous dispersion of particles in the matrix.
· The reduction in porosity led to the coating's improved resistance to corrosion. How do the authors assure the improvement in their system?
· The authors are suggested to write a few sentences to propose some prospective applications of their work in the last paragraph before the Conclusions which are missing in their manuscript.
· How about increasing the concentration of the filler, did the authors try further increasing the filler concentration?? What was the effect ?? And why didn’t add those results in the manuscript?
· The authors are suggested to add some recently published references relevant to their work. Some references are important to understand the progress of polymer-based composite materials and their advantages: Composites Part A, 2022, 153, 106734. Similarly for 2D materials and their composites; Materials Today Communications, 2022, 31, 103858; Composites Science and Technology, 2020, 189, 108022. Please check all the reference formatting again.
· The language expression in the text needs to be carefully checked and revised. There are several serious grammatical mistakes.
Reviewer 3 Report
Jun Tang and coworkers suggested anti-corrosion and wear resistant organic coatings as a solution to overcome rusting and corrosion problems of metallic instruments. The following corrections are suggested to make it a good addition to the field:
1. Quality of image 1 should be improved.
2. Many good publications/literature are published on this topic in the last 2-3 years and recent updates must be added to improve the introduction section.
3. References are not given on Page no 2; para no 2 (Lines: Using GO as a carrier for SiC in the epoxy………… possibility for the self-assembly of SiC and GO.)
3. Usages of Superscript/subscript are recommended in molecular formulas (e.g. Al2O3, Fe3O4,…..etc.) in the reference section.
Round 2
Reviewer 2 Report
The authors have answered all the questions, the reviewer has no further queries. The paper can be accepted for publication.